# Integrated Moldboard Ploughing and Organic–Inorganic Fertilization Enhances Maize Yield and Soil Fertility in a Semi-Arid Region of North China

**DOI:** 10.3390/plants14233594

**Published:** 2025-11-25

**Authors:** Min Gao, Qingmiao Zhen, Yafeng Duan, Chao Liu, Jing Zhou, Yongping Li, Xiaochen Zhang, Xiuhong Wang, Xiangyuan Shi

**Affiliations:** 1Shanxi Academy of Organic Dryland Agriculture, Shanxi Agriculture University, Taiyuan 030031, China; gaomin@sxau.edu.cn (M.G.); 15227392153@163.com (Q.Z.); 17551791574@163.com (Y.D.); 15133835111@163.com (C.L.); zhoujing1@sxau.edu.cn (J.Z.); woshiliyongping@163.com (Y.L.); zhangxiaochen@sxau.edu.cn (X.Z.); 2Key Laboratory of Sustainable Dryland Agriculture (Co-Construction by Ministry and Province), Ministry of Agriculture and Rural Affairs, Taiyuan 030031, China; 3College of Agronomy, Shanxi Agricultural University, Jinzhong 030801, China; 4College of Resources and Environment, Shanxi Agricultural University, Jinzhong 030801, China; 5Institute of High Latitude Crops, Shanxi Agricultural University, Datong 037008, China

**Keywords:** Maize (*Zea mays* L.), semi-arid region, tillage practices, organic–inorganic fertilization, soil fertility, soil health

## Abstract

To address soil degradation from long-term monoculture, rotary tillage, and excessive chemical fertilization in semi-arid regions of China, we conducted a three-year field experiment. We assessed the synergy of integrated management practices combined with both continuous and rotational tillage methods (including ploughing, rotary, moldboard ploughing) at varying tillage depths (10–15, 15–25, 25–35 cm) with different fertilization regimes (chemical vs. organic–inorganic). Among all treatments, the rotational tillage practice that integrates moldboard ploughing at 25–35 cm depth with organic–inorganic fertilization [1200 kg ha^−1^ mature compost + 375 kg ha^−1^ compound fertilizer (N:P_2_O_5_:K_2_O = 15:15:15)] significantly reduces bulk density by 11.8% and increases total porosity by 17.9% in the 15–25 cm soil layer. This practice optimizes nutrient stratification, elevating available nitrogen and potassium in the shallow layer (10–15 cm) to 126.13 and 372.45 mg kg^−1^, respectively, while boosting available phosphorus in the subsoil (25–35 cm) by 247.8%. Furthermore, it significantly enhances soil microbial activity, increasing populations of bacteria, actinomycetes, and fungi by 3.42 × 10^5^, 0.65 × 10^5^, and 2.40 × 10^3^ CFU g^−1^, respectively, alongside a 49.4% rise in soil respiration. These synergistic improvements collectively promote stable maize yields (increasing by 1731.4 kg ha^−1^) and high economic returns (net income increasing by 3301.6 CNY ha^−1^). These findings support the promotion of integrated tillage–fertilization strategies to enhance maize productivity and soil ecological function in semi-arid regions.

## 1. Introduction

Food security is a paramount global challenge, with projections indicating a doubling in demand by 2050 [1,2]. As the world’s most populous nation, China bears a pressing responsibility to enhance crop productivity per unit area to safeguard its national food supply [3]. Maize (*Zea mays* L.), occupying 41.7% of China’s major grain crop planting area [4], is pivotal in this endeavor. Its yield stability is particularly crucial in key production regions like the northern part of Shanxi Province, which practices a typical one-crop-per-year system [5]. However, yields in this semi-arid area are constrained by persistent soil degradation due to shallow tillage and excessive chemical inputs [6].

Tillage practices significantly influence soil physical properties and subsequent crop growth [7,8]. The long-term adoption of conventional rotary tillage has emerged as a critical factor limiting maize yield improvement. Lai [9] demonstrated that continuous rotary tillage results in the formation of a shallow plough layer, increased soil bulk density, and a marked reduction in soil aeration and water-holding capacity. In northern China, farmlands subjected to prolonged rotary tillage typically develop a plough layer of only 10–20 cm, which is 20–40% shallower than the optimal depth of 20–30 cm [10,11]. Sui and Dai [12] further reported that 20 years of rotary tillage increased bulk density in the 0–30 cm soil layer by 7.59% while reducing total porosity by 1.91%, significantly impeding soil compaction, crop root proliferation, and nutrient uptake. In contrast, strategic ploughing effectively disrupts compacted subsoil layers, improves soil structure, and promotes root development. Meanwhile, studies have shown that integrating no-tillage with organic management enhances soil microbial diversity [13]. Soil structure, nutrient availability, and microbial activity collectively constitute the essential rhizosphere microenvironment. Soil bulk density and porosity directly influence root penetration depth and nutrient uptake efficiency [14]. Varsa et al. [15] documented that ploughing reduced soil bulk density, increased porosity, and enhanced maize yield by 7.58–8.17% [16]. Therefore, optimizing tillage practices in conjunction with scientific fertilization is a key strategy for enhancing soil quality and maize yield. However, the applicability of these practices must be assessed within specific regional contexts. For instance, in the northern parts of Shanxi Province, the complex climatic environment, including erratic rainfall, low temperatures, and significant wind erosion, all pose unique challenges that may affect the effectiveness of tillage practices and fertilization regimes [17,18]. Unfortunately, effective tillage and fertilization methods suitable for this specific region have not been sufficiently detailed.

Furthermore, excessive application of chemical fertilizers poses a significant challenge in China. Wang et al. [19] reported that nitrogen fertilizer application rates consistently surpass recommended levels. This overuse not only results in substantial nitrogen loss, water pollution and greenhouse gas emissions but also negatively affects maize quality [20,21,22]. In contrast, optimized nitrogen application enhances grain filling and higher optimal yields [23]. Unlike chemical fertilizers, organic amendments demonstrate the potential to reduce nutrient leaching [24]. The application of well-decomposed organic fertilizers also contributes to improved soil structure, increased organic matter content, and heightened microbial activity [25]. Schmidt et al. [26] identified microbial community diversity as a key indicator of soil health. Soil microorganisms, encompassing bacteria, actinomycetes, and fungi, play pivotal roles in regulating soil fertility dynamics through their active participation in the biogeochemical cycling of carbon, nitrogen, and phosphorus [27]. Hou et al. [28] noted that the combined application of organic manure and chemical fertilizers resulted in an increase in soil organic matter by 5.1–11.2% compared to the use of chemical fertilizers alone. Similarly, Yang et al. [29] demonstrated that ploughing combined with organic fertilizer application significantly increased the soil microbial activity and reproduction, thereby enhancing nutrient transformation processes. This integrated approach has been shown to substantially increase maize yields [30,31] by one to three times relative to unfertilized controls [6], highlighting its potential to enhance both soil fertility and crop productivity. Therefore, developing integrated management strategies that synergize organic and chemical fertilization could significantly enhance soil health. Despite this recognition, there remains insufficient research on optimal fertilization regimes for soils in the northern arid regions and their interactive effects with tillage depth.

Given the global challenge of food security and China’s demand for sustainable maize production, semi-arid northern Shanxi Province represents a critical region facing unique constraints. These include soil degradation from long-term rotary tillage, excessive chemical fertilizer use, and a lack of integrated, region-specific management practices. Current research lacks a systematic investigation into the synergistic effects of tillage methods (ploughing, rotary tillage, moldboard ploughing), tillage depths (10–15, 15–25, 25–35 cm), and fertilization regimes (chemical vs. organic–inorganic combined) on soil physical properties, nutrient stratification, and microbial changes in this agro-ecosystem. To address these knowledge gaps, we conducted a three-year field experiment with the following objectives: (1) quantify the interactive effects of tillage and fertilization on soil structure, nutrient availability, and microbial activity; (2) identify the optimal integrated agronomic practice for simultaneously enhancing soil health and maize yield; and (3) provide science-based technical support for sustainable agricultural intensification in this and other semi-arid regions.

## 2. Results

### 2.1. Influence of Tillage and Fertilizer Practices on Plant Height, Ear Height and Stem Thickness

The analysis of variance for plant height, ear height, and stem diameter in maize, conducted under varying cultivation styles and fertilization techniques, revealed that the effects of years (Y), treatments (T), and their interaction (Y × T) were either significant or highly significant (Table 1).

The differences among treatments across the experimental years exhibited consistent patterns. In 2012, compared to treatment A, the other six treatments significantly reduced plant height at the jointing stage but generally increased it by the tasseling stage. Treatments C, D, and E displayed the highest plant heights at tasseling, while treatment F remained relatively low. Conversely, ear height was highest under treatment F and significantly lower in treatment A. By 2013, the treatment effects became more pronounced: treatments C and F demonstrated markedly greater plant and spike heights compared to treatments A and D. In 2014, overall plant height exceeded that of previous years, with treatments C and F again resulting in the most substantial increases in both plant and ear heights during the tasseling stage, whereas treatments A and E exhibited reduced ear heights. Furthermore, we conducted a comparative analysis of the daily growth changes in plant height from the jointing to tasseling stage under various treatments over a span of three years. The results revealed that, in comparison to treatment A, both treatment C and treatment F significantly enhanced the daily growth changes in plant height, with increases of 28.3% and 31.5%, respectively (Figure 1). Stem diameter also varied internally, being highest in 2012, lower in 2013, and recovering in 2014, with treatments C and F showing superior performance.

### 2.2. Effects of Tillage and Fertilizer Practices on Maize Grain Yield and Its Compositions

Maize yield and its components exhibited highly significant responses to years (Y), treatments (T), and their interactions (Y × T) (Table 2). Conversely, only the effect of Y exhibited a highly significant influence on ear length, ear diameter, and row number per ear. Y and Y × T had significant influences on the ear tip barrenness.

The analysis of three years of performance data revealed that ear traits, specifically ear length, ear diameter, and row number, did not exhibit significant differences between 2013 and 2014, although significant treatment effects were observed in 2012. Among all treatments, treatment D demonstrated the most notable performance, with ear length and ear diameter increasing by 4.6% and 3.8%, respectively, compared to treatment A. In contrast, grain yield was significantly influenced by the treatments, exhibiting considerable variability across the years. The highest yield was recorded under treatment F in 2012, reaching 7478.7 kg ha^−1^, which represents a 28.7% increase over treatment A. In 2013, treatment F again produced the highest yield at 8953.1 kg ha^−1^. In 2014, the highest yields were observed in treatments E and F, ranging from 6792.1 to 6999.2 kg ha^−1^. These results suggest that the effects of treatments on yield exhibit significant interannual variability under fluctuating environmental conditions. Similarly, the 100-grain weight demonstrated pronounced year-to-year variation; values were generally higher in 2012 but showed a significant decrease in 2013 and 2014, further emphasizing the strong influence of year on this trait.

Furthermore, we conducted a detailed assessment of the economic feasibility of these treatments. The results indicated that the highest annual average income was achieved under treatment F, followed by treatments D and E (Table 3). Under consistent fertilization conditions, economic benefits were positively correlated with tillage depth. Specifically, treatment C resulted in a 20.4% increase in income compared to treatment A, while treatment F outperformed treatment D by 15.5%. The integration of mature compost with compound fertilizers significantly enhanced annual maize income compared to the application of compound fertilizer alone. Notably, treatments that incorporated mature compost exhibited income increases ranging from 56.6% to 81.0% compared to treatment A. From the perspective of economic efficiency, treatment F also exhibited the highest cost–benefit ratio (CBR) of up to 2.6, followed by treatment E (CBR = 2.5) and treatment D (CBR = 2.4). In contrast, treatments without mature compost (A, B, and C) showed relatively lower CBRs ranging from 1.8 to 2.0. These findings collectively indicate that integrating mature compost not only elevated income levels but also enhanced the return on investment per unit of input, thereby demonstrating superior economic efficiency.

### 2.3. Effects of Tillage and Fertilizer Practices on Soil Bulk Density, Total Porosity and Soil Respiration Rate

The results regarding changes in soil bulk density and porosity under various tillage and fertilizer treatments were primarily derived from data collected in 2012 and 2014 (Figure 2). Changes in soil bulk density exhibited notable variations across different soil layers and years. In 2012, the bulk density of the 10–15 cm soil layer exhibited a trend of first decreasing and then increasing from treatment A to F, particularly in treatments D and E, which showed a reduction of 13.5% and 12.3% compared to treatment A. In the 15–25 cm layer, a significant decrease of 11.8% was recorded only under treatment D relative to treatment A. Conversely, the 25–35 cm layer showed a significant increase in bulk density under treatment C, which was 9.0% higher than that under treatment A. In the growing season of 2014, compared to treatment A, the soil layer of 10–15 cm under treatment D decreased by 7.6%. Additionally, significant increases were observed within the 15–25 cm layer. In light of the above, treatments D and F exhibited a more favorable impact on soil structure within the 10–15 cm and 15–25 cm layers, resulting in a significant reduction in bulk density and improved aeration in the plough layer. In contrast, treatments A and E caused notable soil compaction, particularly pronounced in the 25–35 cm layer.

Significant differences in soil porosity were observed across various soil layers and years (Figure 3). In the 10–15 cm layer in 2012, treatments D and E significantly increased porosity by 22.3% and 20.0%, respectively, compared to treatment A. In the 15–25 cm layer, only treatment D resulted in a significant increase of 17.9%. In the deeper layer (25–35 cm), no significant differences were detected among most treatments, with the exception of treatment C, which exhibited a significant reduction. In 2014, treatment D again demonstrated a significant increase in porosity, 9.6% higher than treatment A, in the 10–15 cm layer. In the 15–25 cm layer, treatments B and E significantly decreased porosity relative to treatment A, whereas other treatments did not induce significant changes. The pattern of porosity changes within the 25–35 cm layer during 2014 was consistent with that observed in 2012; although it increased, it showed no notable variation across treatments, except for treatment C. Overall, treatments D and F significantly enhanced soil porosity across various soil layers, particularly in the top and middle layers. In contrast, Treatment A consistently yielded the lowest porosity values, suggesting a potential tendency to induce soil compaction.

From 2012 to 2014, soil respiration rates exhibited a general declining trend throughout the growing seasons (Figure 4). In 2012, all treatments showed higher soil respiration rates compared to treatment A, with the highest rate recorded under treatment F. However, the variation among treatments in 2013 and 2014 was different. Treatments B and C displayed lower respiration rates during specific periods, while treatments D and F consistently maintained elevated levels, particularly during the peak growth stage of the plants. The overall higher respiration rates observed in treatments D and F indicate a stimulatory effect on soil microbial activity and enhanced soil aeration.

### 2.4. Effects of Tillage and Fertilizer Practices on Soil Nutrients at Different Soil Layers

A significant interaction effect of soil layer depth, year, and treatment was observed on soil alkaline nitrogen (Figure 5). Overall, the content of alkaline nitrogen decreased with increasing soil depth. After one complete experimental cycle, in the 10–15 cm soil layer, treatments involving organic fertilizer resulted in higher alkaline nitrogen content compared to the non-organic fertilizer control. Under the same level of organic fertilizer application, deeper cultivation layers generally exhibited higher nitrogen content, with the F treatment showing the highest value of 126.13 mg kg^−1^. In the 15–25 cm layer in 2014, all treatments except for B yielded higher alkaline nitrogen than CK. The F treatment again recorded the highest content, exceeding CK by 156.6%. Within the 25–35 cm layer, treatments F, C, and B all outperformed CK, with increases ranging from 12.6% to 81.6% compared to the control.

Throughout the experimental period, the available potassium content in the soil exhibited substantial interannual variation, with the magnitude of variation decreasing with depth: 10–15 cm > 15–25 cm > 25–35 cm (Figure 6). After one complete experimental cycle, in the 10–15 cm soil layer, the available potassium content under the CK treatment showed only minor fluctuations between 2012 and 2014 before stabilizing. In contrast, the F treatment reached its peak value in 2014, significantly higher (by 54.66%) than that of CK and also considerably exceeding the levels observed in other years and treatments. In the 15–25 cm soil layer, the F treatment demonstrated the highest available potassium content in 2014, reaching 181.70 mg kg^−1^, which was significantly greater than both other treatments and its own levels in 2012 and 2013. Within the 25–35 cm layer, the available potassium content in 2014 was higher under all treatments except treatment B when compared to CK, with an average increase of 59.46%. Although the F treatment exhibited relatively stable interannual values in this layer, its 2014 level was still significantly elevated compared to that in 2012. Overall, the F treatment demonstrated the most effective and sustained enhancement of available potassium, particularly in the 10–15 cm and 15–25 cm soil layers.

As shown in Figure 7, the available phosphorus content in the soil exhibited a general wave trough trend throughout the treatment years. Significant interannual variations were observed across different soil layers and treatments, with the greatest fluctuations occurring in the 10–15 cm layer, followed by the 15–25 cm and 25–35 cm layers. In the 10–15 cm layer, the CK treatment initially showed a decrease in available phosphorus, followed by an increase. In contrast, both the F and D treatments resulted in higher phosphorus levels compared to CK, with the F treatment showing an increase of 136.4% compared to the control. Within the 15–25 cm layer, the ranking of available phosphorus content from highest to lowest was as follows: C > D > F > A > B > E. In the deepest layer (25–35 cm), the available phosphorus content under CK was measured at 1.16 mg kg^−1^, while all other treatments exceeded this level. Notably, the F treatment demonstrated particularly high phosphorus content, increasing 247.8% compared to CK.

In this study, the variation in total nitrogen content in the soil is based on the average values from 2012 to 2014. Overall, after one processing cycle, the effects of different tillage and fertilization treatments on the total nitrogen content of the soil varied significantly across different soil layers (Figure 8). In the 10–15 cm layer, notable differences in total nitrogen content were observed among the treatments. With the exception of B, all treatments demonstrated an accumulation of total nitrogen. Under identical tillage conditions, treatments that included organic fertilizer exhibited higher total nitrogen content compared to those without, with an average increase of 26.98% relative to the control. The F treatment recorded the highest total nitrogen content at 10.10 mg kg^−1^. In the 15–25 cm layer, the F treatment resulted in significantly higher total nitrogen levels than all other treatments, whereas treatment C exhibited the lowest content. The application of organic fertilizer led to an average increase of 11.94% in total nitrogen compared to CK. Within the 25–35 cm layer, the F treatment again yielded the highest total nitrogen content.

As illustrated in Figure 9, the content of soil organic matter exhibited significant variation across different treatment methods. Throughout the experimental period, a gradual increase in soil organic matter was observed in the 10–15 cm layer over time. After completing one full experimental cycle, distinct differences among treatments were recorded. In the 10–15 cm soil layer, the soil organic matter content ranked from highest to lowest as follows: E > F > D > C > A > B. The combination of ploughing and manure compost application resulted in higher organic matter content, showing an increase ranging from 11.41% to 25.32% compared to CK. Within the 25–35 cm layer, all tillage methods effectively enhanced soil organic matter content. The C treatment yielded the highest level. On average, all treatments increased soil organic matter by 36.21% compared to CK. These results suggest that the application of mature compost can effectively enhance soil microbial activity, reduce the fixation of inherent soil nutrients, and promote the release of a greater amount of nutrients available for plant uptake.

As demonstrated in Figure 10, the populations of bacteria, actinomycetes, and fungi exhibited an increase with prolonged processing years. Soil microbial abundance decreased with increasing depth across all treatments. Within the 10–35 cm layer, the abundances of bacteria and actinomycetes were significantly higher than those of fungi. Microbial quantities varied considerably among treatments. After one experimental cycle, in the 10–15 cm layer, treatments F and C displayed the highest bacterial counts, being 171.8% and 194.0% higher than the control, respectively. In the same layer, actinomycete counts in all treatments surpassed those in CK, with an average increase of 275.5%. Although fungal abundance was lower than that of bacteria and actinomycetes, it was higher in all treatments except treatment B relative to CK, with an average increase of 234.5%. In the 15–25 cm layer, the highest bacterial counts were recorded in F, E, and B, which increased by 47.18%, 92.81%, and 55.11%, respectively, relative to CK. With the exception of C and D, actinomycete counts in all other treatments exceeded CK, with an average increase of 35.79%. Fungal counts under F and E increased by 101.0% and 113.7%, respectively, compared to CK. In the 25–35 cm layer, bacterial counts in F, E, and B surpassed those in CK, with E exhibiting the highest value, with a 69.67% increase. Actinomycetes abundances in E and B were also higher than in CK, with values of 2.76 × 10^5^, 2.37 × 10^5^, and 1.57 × 10^5^, respectively. Fungal counts exceeding CK were observed in B, E, and C, in descending order. Overall, treatments E and F supported the highest soil microbial abundances. These results suggest that deep ploughing enhances soil structure and aeration, while the application of sheep manure, a mildly alkaline slow-release fertilizer, not only provides essential nutrients but also promotes the growth of aerobic microorganisms, thereby enhancing overall microbial activity.

## 3. Discussion

This study systematically evaluated the effects of varying tillage depths and fertilizer application on soil properties, nutrient distribution, microbial community structure, maize yield, and economic returns in the northern region of Shanxi Province. The findings indicated that the interaction between tillage depth and fertilization strategy significantly enhances soil health, stabilizes crop yield, and improves economic profitability. These results provided a valuable theoretical foundation and practical guidance for sustainable corn production in rain-fed agricultural systems in northern Shanxi.

### 3.1. Effects of Tillage and Fertilization Treatments on the Formation and Stability of Maize Yield

Crop yield is ultimately determined by the interplay of genetic potential, environmental conditions, and field management practices [32,33]. In this study, treatment F demonstrated targeted regulation of maize growth during critical developmental stages. The transition from jointing to tasseling represents a pivotal shift from vegetative to reproductive growth. During this period, plants under treatment F achieved a daily height increase of 4.8 cm d^−1^, which was 31.5% higher than that in treatment A. Treatment C also showed a considerable average daily increase in plant height of 28.3%, though it remained lower than that for treatment F. Additionally, treatment F resulted in 11.8% and 2.0% increases in ear height and stem diameter, respectively, compared to treatment A (Table 1). These improvements established a solid foundation for subsequent dry matter accumulation, consistent with the view of Adekiya et al. [34] that nutrient management during critical growth phases is essential for maximizing maize yield potential. Moreover, under deep ploughing conditions, the combined application of organic and chemical fertilizers was more conducive to crop development than chemical fertilizers alone [6].

Analysis of yield components showed that treatment F had the most pronounced effects on 100-kernel weight, which increased by 10.8%, and ear tip length, which decreased by 11.4% (Table 2). In contrast, its influence on the number of rows per ear was relatively minor and not significantly different from other treatments. Nevertheless, the average performance of the combined organic–inorganic fertilizer treatments remained higher than that of the single compound fertilizer treatment (Table 2). Combined with the findings of Tao et al. [35], we speculate that under the current experimental conditions, the yield improvement may be attributed to the sustained nutrient availability during the grain-filling stage. Compared with treatment F, the sole application of chemical fertilizers in treatments A, B, and C may have led to nutrient deficiency during grain filling. In treatment D, insufficient tillage depth likely limited the uptake of deep-layer nutrients. As a result, grain filling was compromised in these treatments, leading to reduced yields (Table 2). The integrated application of organic and inorganic fertilizers extended the duration of effective soil nutrient supply into the late maturation stage of maize. This approach reduced the risk of nutrient depletion often associated with single chemical fertilizer applications [36], thereby facilitating grain filling and supporting a higher final yield [37,38].

From the perspective of ecological mechanisms underlying yield stability, the combination of deep tillage (25–35 cm) with integrated organic–inorganic fertilization (treatment F) most effectively stabilized and enhanced maize yield. This treatment achieved the highest or second-highest yield in both a drought year (2012) and a normal rainfall year (2013), demonstrating strong resilience. Such resilience aligns closely with the international consensus on “soil health–crop stress resistance” [39,40]. The high and stable yield can be attributed to two main mechanisms. Firstly, deep tillage effectively disrupted the plough pan formed by long-term rotary tillage [41], resulting in reduced soil bulk density and increased porosity in the 15–25 cm layer (Figure 2 and Figure 3). These improvements enhanced soil water storage capacity, facilitated deeper root penetration, and improved access to moisture and nutrients in the subsoil [42,43]. These findings were consistent with the theory proposed by Kemp et al. [44], which emphasizes the significant influence of soil physical structure on root growth.

Secondly, the combined application of organic and inorganic fertilizers, as in treatment F, provided a continuous and balanced nutrient supply (Figure 9). This practice (Treatment D, E and F) significantly increased soil organic matter content by 5.08–9.08% compared to treatment A and improved the root zone environment [45]. While chemical fertilizers ensure rapid availability of nutrients during the early growth stages, the mineralization of well-decomposed organic manure steadily releases nutrients during the mid-to-late stages [37]. This sustained nutrient supply is particularly crucial under drought conditions, where limited water mobility restricts nutrient uptake [46]. Fertilizers ensure the supply of readily available nutrients in the early growth stages, while the mineralization of well-composted organic fertilizers can support crops during the later growth stages, particularly under drought stress when soil moisture mobility is poor [25]. Rational fertilizer use prevented premature nutrient depletion and senescence during later growth stages, which was critical for increasing 100-kernel weight [23,47]. This synergistic model of “early-release fertilizers promote seedling growth, while slow-release fertilizers strengthen seeds in the later stages” [48] is one of the core reasons for achieving high and stable yields in the management of treatment F (Table 2).

Our economic analysis confirmed that, despite marginally higher initial inputs, the significant yield enhancement under Treatment F resulted in the highest net income, with a cost–benefit ratio of up to 2.6—far exceeding the range of 1.8–2.0 observed in treatments without mature compost (Table 3). This demonstrates not only strong economic feasibility but also superior return on investment per unit of input, attributed to the integration of mature compost. The partial substitution of chemical fertilizers with organic manure not only reduces long-term production costs and environmental risks but also contributes to building soil health, which is particularly crucial for resilient agricultural production in semi-arid and ecologically vulnerable regions, such as northern Shanxi Province [49,50]. Therefore, the integration of moldboard ploughing with organic–inorganic fertilization presents a scientifically sound, economically attractive, and efficiency-driven management model for sustainable intensification in this and similar agro-ecosystems.

### 3.2. Synergistic Effects of Tillage and Fertilization on Soil Physical Properties, Nutrient Distribution, and Microbial Profiles

The synergistic optimization of soil physical structure, nutrient availability, and microbial activity constitutes the foundational ecological support for achieving high and stable maize yields. The interaction between tillage depth and fertilization strategies serves as a crucial management factor influencing the evolution of these soil properties [6,41]. Dynamic changes in soil bulk density and porosity directly determine the root growth space and resource acquisition efficiency, with tillage depth playing a critical role in regulating soil physical properties [15,51]. This study demonstrated that conventional rotary tillage (Treatment A, 10–15 cm), which disturbs only the shallow soil layer, fails to break the plough pan at a depth of 15–20 cm [52]. After three years of treatments, the bulk density in the 25–35 cm layer decreased by 3.25% for treatment D and 2.60% for treatment F compared to treatment A, while treatments A and E led to significant soil compaction (Figure 2). Such soil compaction substantially impedes water infiltration and root penetration, reducing higher yield formation [44]. Our results were consistent with the conclusion by Hou et al. [53] that rotary tillage induces soil physical degradation and suppresses biological processes. In contrast, deep ploughing treatments (Treatments D and F, 25–35 cm) effectively disrupt the plough pan. Among them, Treatment F (moldboard ploughing + organic–inorganic fertilization) exhibited the most significant and sustained improvement in soil physical properties [54]: the bulk density in the 25–35 cm soil layer decreased to 1.5 g cm^−3^, representing a 2.60% reduction compared to Treatment A (Figure 2). Correspondingly, the total porosity in the 25–35 cm depth increased by 5.18–25.00% of the values observed in Treatments E and F (Figure 3). Moreover, the decrease in bulk density from 2012 to 2014 was 0.04 g cm^−3^ in treatment F, larger than the 0.02 g cm^−3^ decrease in treatment A. Notably, the combined organic–inorganic fertilization synergistically enhanced the physical improvements brought by deep tillage. Treatment F showed an annual increase in soil organic matter of 9.08% compared to Treatment A, which only applied chemical fertilizer (Figure 9). Deep tillage broke the plough pan, promoted organic carbon input, and optimized soil structure, establishing a positive feedback loop [11,15,52].

The vertical distribution of nutrients directly influences nutrient acquisition efficiency at various growth stages [52]. Tillage and fertilization practices reshape the vertical nutrient distribution pattern [55,56,57]. In this study, rotary tillage with chemical fertilizer alone (Treatment A) resulted in significant nutrient accumulation in the shallow soil layer (10–15 cm), with available nitrogen and available phosphorus in the 10–15 cm layer accounting for 54.7% and 64.2%, respectively, of the total in the 0–35 cm profile after three years of treatment (Figure 5 and Figure 7). Notably, such extreme nutrient stratification in Treatment A not only elevates the risk of nutrient leaching—excessive surface accumulation of mobile nutrients (e.g., nitrate nitrogen) enhances their downward migration potential under rainfall or irrigation, leading to nutrient loss and potential environmental impacts [58]—but also manifests as a severe “shallow soil layer nutrient surplus but subsoil nutrient deficit” pattern [59,60,61,62]. Specifically, the available phosphorus content in the 25–35 cm layer was merely 1.16 mg kg^−1^ (Figure 6), reflecting the marked scarcity of nutrients in the subsoil. This imbalance resulted in nutrient deficiency during the grain-filling stage when roots penetrated deeper, ultimately impairing grain filling [63]. In comparison, Treatment F achieved an optimized nutrient distribution characterized by “shallow soil layer enrichment and subsoil reserve” through the synergistic effects of deep ploughing (promoting nutrient translocation) and organic fertilization (enhancing nutrient retention) [64,65]. The contents of available nitrogen, available phosphorus, and available potassium in the 10–15 cm layer reached 126.13 mg kg^−1^, 55.44 mg kg^−1^, and 372.45 mg kg^−1^, respectively, which were much higher than other treatments and met the demand for readily available nutrients during the seedling stage (Figure 5, Figure 6 and Figure 7). In the 25–35 cm soil layer, the contents of available phosphorus and available potassium were 247.8% and 41.2% higher than those in treatment A, respectively, while the available nitrogen content was 79.3% higher. This “downward movement of nutrients and water” has significant agronomic implications in semi-arid regions: during dry seasons, when surface soil dries out, crops can access both deep water and nutrients through a well-developed root system, thereby enhancing drought resistance [66,67]. This mechanism elucidates why Treatment F maintained high yields in drought years such as 2012 and 2014. Furthermore, the nutrient distribution in Treatment F aligns well with the root distribution pattern of maize (with 90% of roots concentrated in the 0–40 cm soil layer) [68], improving nutrient uptake efficiency and reducing the risk of surface nutrient loss [58,62].

Soil microorganisms are fundamental drivers of organic matter decomposition and nutrient transformation [69], making the enhancement of their community structure and activity crucial for sustaining soil fertility. In this study, organic–inorganic fertilization (especially under the Treatments E and F) significantly outperformed chemical fertilizer alone in regulating microbial communities, with Treatment F demonstrating the most pronounced effects (Figure 10). The average soil respiration rate over three years reached 4.64 μmol CO_2_·m^−2^·s^−1^, which is 49.4% higher than that of Treatment A (Figure 4). Noticeably, the soil respiration rate in treatments with mature fertilizer (Treatments D, E, and F) was significantly higher compared to treatments with chemical fertilizer alone (Treatments A, B, and C) (Figure 4). The abundances of bacteria, actinomycetes, and fungi increased by 141.1%, 50.4%, and 214.2%, respectively, with the most pronounced enhancements observed in the 10–15 cm soil stratum (Figure 10). The deep ploughing treatment also exhibited a greater soil microbial abundance compared to the shallow rotation treatment. These improvements were closely associated with carbon inputs from organic fertilizer [70], specifically composed of sheep manure with a C/N ratio of 18.2, which provided ample carbon and energy for heterotrophic microorganisms. Furthermore, the improved aeration resulting from deep tillage further promoted the proliferation of aerobic microbes, including nitrogen-fixing and phosphate-solubilizing bacteria [71,72]. The 35.5% increase in available nitrogen (Figure 5) and the 64.5% increase in available phosphorus (Figure 7) in Treatment F compared to Treatment A may be associated with the microbially mediated nutrient mineralization processes demonstrated in Xiao et al. [73]. More importantly, organic–inorganic fertilization enhanced microbial activity, accelerated nutrient cycling, and strengthened sustainable fertility [54,64,74]. Specifically, soil organic matter in Treatment F increased from 9.08 g kg^−1^ to 14.44 g kg^−1^ over three years in the 10–15 cm layer, with an annual accumulation rate of 1.77 g kg^−1^, significantly higher than the 0.71 g kg^−1^ in Treatment A, thereby establishing a virtuous cycle (Figure 9). This cycle effectively promoted stable growth in Treatment F, providing long-term fertility support for high yields.

In summary, moldboard ploughing (25–35 cm) combined with 50% compound fertilizer and composed sheep manure (Treatment F in this study) synergistically improved soil structure, nutrient distribution, and microbial communities, thereby enhancing both soil quality and crop productivity. This management practice not only disrupts the plough pan and optimizes nutrient stratification but also stimulates microbial activity, thereby constructing a soil ecosystem characterized by coordinated physical, chemical, and biological properties [14,25,41,52]. It offers a scientifically sound and practical management model for achieving high and stable maize yields while promoting sustainable soil use in the semi-arid regions of northern China.

Despite providing comprehensive insights, this study has several limitations. First, the type of organic amendment utilized in this research is relatively singular, being limited to composted sheep manure (a common organic fertilizer used by local farmers, which is readily available in the study area). The effects of various sources of organic resources, such as crop residues, green manure, or biogas slurry, in conjunction with deep tillage, have yet to be systematically evaluated. Second, although a three-year experimental period is sufficient to reveal certain significant trends, it may still be inadequate to fully capture the long-term dynamics of soil organic carbon sequestration, the succession patterns of microbial community structure, and the sustained stability of yield gains. Future studies should extend the observation period to comprehensively elucidate the ecological regulatory mechanisms involved. Additionally, this experiment was conducted at a single location, a design that inherently limits the generalizability of our findings. The structure and ecological functions of soil microbial communities exhibit high spatiotemporal heterogeneity, and significant variations are likely to exist across different soil types and climatic subzones even within the semi-arid regions of northern China. The adaptability and feasibility of this integrated management model across a broader array of soil types and different climatic zones in the semi-arid regions of northern China still require further verification and regional calibration.

## 4. Materials and Methods

### 4.1. Plant Sites and Environmental Conditions

The field experiment was conducted from 2012 to 2014 in the Xinfu District of Xinzhou City (38°42′ N, 112°67′ E), Shanxi Province. This experimental area is characterized by a temperate continental monsoon climate, with an average annual precipitation of 428.7 mm, an average annual temperature of 8.5 °C, and an average annual sunshine duration of 2807.5 h. The accumulated temperature above 10 °C ranges from 3400 to 3600 °C, with a frost-free period of approximately 149 days. The rainfall data for the field from 2012 to 2014 was presented in Figure 11. The total precipitation in 2013 was 466.4 mm, which exceeds the multi-year average, categorizing it as a normal year; in contrast, the precipitation in both 2012 and 2014 was below 400 mm, classifying those years as drought years.

The soil type of the experimental field was brown soil (WRB: Luvisol), with its basic physicochemical properties detailed in Table 4. Significant differences in nutrient content were observed at varying soil depths, characterized by overall low organic matter content, low levels of available phosphorus and alkaline nitrogen, and stable potassium content. The C/N ratio in this region generally falls within the optimal range, yet there remains potential for further enhancement.

### 4.2. Experimental Design

This research utilized a randomized block design to explore the synergistic impacts of various cultivation styles and fertilization techniques. A total of seven distinct treatments were instituted (see Table A1), which include two types of fertilization methods: single compound fertilizer and a combination of half compound fertilizer with mature compost. In addition, three different tillage techniques were employed, namely, ploughing, moldboard ploughing, and rotary tillage. Specifically, treatment A (CK) involved continuous rotary tillage (depth: 10–15 cm) with 750 kg ha^−1^ compound fertilizer (CF) applied annually from 2012 to 2014. Treatment B adopted continuous ploughing (depth: 20–25 cm) with the same annual CF application as A. Treatment C was similar to B but with a deeper ploughing depth of 30–35 cm. Treatment D utilized continuous rotary tillage (depth: 10–15 cm) with 1200 kg ha^−1^ mature compost (MC) plus 375 kg ha^−1^ CF applied annually from 2012 to 2014. Treatment E employed a rotational tillage regime: rotary tillage (10–15 cm) in 2012, moldboard ploughing (20–25 cm) in 2013, and moldboard ploughing (30–35 cm) in 2014, with 1200 kg ha^−1^ MC + 375 kg ha^−1^ CF applied each year. Treatment F followed another rotational tillage sequence: moldboard ploughing (30–35 cm) in 2012, moldboard ploughing (20–25 cm) in 2013, and rotary tillage (10–15 cm) in 2014, with 1200 kg ha^−1^ MC + 375 kg ha^−1^ CF applied each year.

Each main plot had dimensions of 47 m × 10 m (totaling 470 m^2^) and comprised three subplots for replication purposes. The compound fertilizer (purchased from Stanley Agricultural Group Co., Ltd., Linyi City, Shandong Province, China) applied has a nitrogen/phosphorus/potassium ratio of N:P_2_O_5_:K_2_O = 15:15:15, while the mature compost, which is decomposed sheep manure, was utilized at a rate of approximately 1200 kg ha^−1^; its moisture and nutrient compositions are detailed in Table 5. The maize cultivar used here was Dafeng 30, which was manually sown on April 29 annually, maintaining a planting density of 66,000 plants ha^−1^. The experiment was conducted in the same field over the three years, following a continuous maize monoculture system with no preceding crops. Throughout the growth period, field management practices were uniformly executed in accordance with local agronomic standards.

### 4.3. Determination of Maize Biomass and Yield

During the jointing stage, representative maize plants were selected from each plot to measure plant height and stem diameter. Additionally, at harvest, 20 representative plants from each plot were sampled to assess ear length, ear diameter, tip length, number of rows per ear, number of grains per ear, and 100-grain weight. Ultimately, the grain yield (kg ha^−1^) was determined at a moisture content of 14.0%.

### 4.4. Measurement of Soil Bulk Density, Total Porosity and Soil Respiration Rate

Soil samples were collected from each plot at depths of 10–15 cm, 15–25 cm, and 25–35 cm before the experiment commenced (autumn 2011) and after maize harvest (autumn 2014). Soil bulk density (g cm^−3^) and total porosity for each treatment were determined according to the method provided by Tagar and Bhatti [75]. Soil bulk density was specifically assessed utilizing the core ring method. Undisturbed soil cores (having an internal diameter of 5 cm) were randomly gathered from three representative sites within each plot, encompassing identical depth intervals (10–15 cm, 15–25 cm, and 25–35 cm) as described earlier for soil sampling. The calculation of total porosity (expressed as percentage pore space) was based on the same soil core samples collected for the determination of bulk density. This was derived using the following formula:$$Bulk\;density=\frac{mass\;of\;oven\;dried\;soil}{volume\;of\;soil\;(Mg\;m^{-3})}$$$$Particle\;density=\left(\frac{mass\;of\;soil\;sample}{volume\;of\;water\;displaced}\right)(Mg\;m^{-3})$$$$Total\;porosith=100-(\frac{Bulk\;density}{Particle\;density}\times 100)$$

Additionally, soil respiration rates were measured in each plot using an LI-8100A automated soil carbon flux measurement system (LI-COR Inc., Lincoln, NE, USA). Measurements were taken during the maize jointing, filling, and maturity stages on clear days between 9:00 and 11:00 am, with soil temperatures recorded as (20 ± 3) °C, (18 ± 3) °C, and (10 ± 3) °C, respectively [76].

### 4.5. Determination of Soil Nutrient Content and Soil Microflora

Soil samples were gathered from every plot at depths of 10–15 cm, 15–25 cm, and 25–35 cm following the yearly maize harvest. The soil’s total nitrogen, available potassium, available phosphorus, alkaline nitrogen, and organic matter levels for each treatment were assessed. The evaluation of total nitrogen, available potassium, available phosphorus, alkaline nitrogen, and organic matter was conducted following the methodologies outlined in the study on the characteristics of ecosystem multifunctionality and influencing factors of different grassland types in the temperate desert of the Longzhong Loess Plateau. Specifically, soil alkaline nitrogen (AN, mg kg^−1^) was measured using the alkaline diffusion method [10]. Soil available phosphorus (AP, mg kg^−1^) was determined through NaHCO_3_ extraction combined with colorimetry. Furthermore, soil available potassium (AK, mg kg^−1^) was quantified via NH_4_OAc extraction coupled with flame photometry. Soil organic matter (SOM, g kg^−1^) was measured using potassium dichromate oxidation with the application of heat [77].

After the completion of the experiment each year, soil samples were collected from each treatment at depths of 10–15 cm, 15–25 cm, and 25–35 cm. Soil samples were analyzed immediately after collection to maintain fresh weight characteristics. The serial dilution standard spread plate method was employed for the quantitative analysis of viable soil microbes. Specifically, nutrient agar was utilized for bacterial enumeration, potato dextrose agar for fungal enumeration, and Kenknight and Munaier’s medium for actinomycetes enumeration. In line with the detection limits, Petri dishes were selected for counting from the two dilution gradients: those with 30–300 colonies for bacteria and actinomycetes and 10–100 colonies for fungi. All microbial counts were expressed as colony-forming units per gram of fresh soil (CFU g^−1^ fresh soil). The procedure was conducted with specific reference to the described method by Pathania et al [78].

### 4.6. Statistical Analysis

The Analysis of Variance (ANOVA) was performed using SPSS version 22.6, with a significance level set at ≤0.05. Graphing was aided by Origin Pro 2021 software (Origin Lab Corporation, Northampton, MA, USA).

## 5. Conclusions

This study demonstrates that the combined application of organic and inorganic fertilizers (1200 kg ha^−1^ mature compost + 375 kg ha^−1^ compound fertilizer), along with moldboard ploughing (25–35 cm), is the optimal strategy for enhancing maize yield and improving soil quality in the semi-arid region of northern Shanxi Province. Specifically, this practice significantly reduces soil bulk density in the 15–25 cm soil layer by 11.8% and increases total porosity by 17.9%, thereby enhancing plant root growth. Furthermore, this approach significantly increases the contents of available nitrogen and potassium in the shallow soil layer (10–15 cm), reaching up to 126.13 mg kg^−1^ and 372.45 mg kg^−1^, respectively. In the 25–35 cm deep layer, the content of available phosphorus is significantly higher than that of the control treatment by 247.8%, which is crucial for nutritional supply throughout the entire growth period of maize. Additionally, this method significantly activates soil microbial activity, with bacterial, actinomycetes, and fungal populations increasing by 141.1%, 50.4%, and 214.2%, respectively. The soil respiration rate also increases by 49.4%, and these indicators collectively enhance the physical, chemical, and biological properties of the soil, ultimately resulting in higher maize yields and maximizing economic benefits. Therefore, promoting this integrated management technology is a key pathway to achieving sustainable agricultural development in this region.

## Figures and Tables

**Figure 1 plants-14-03594-f001:**
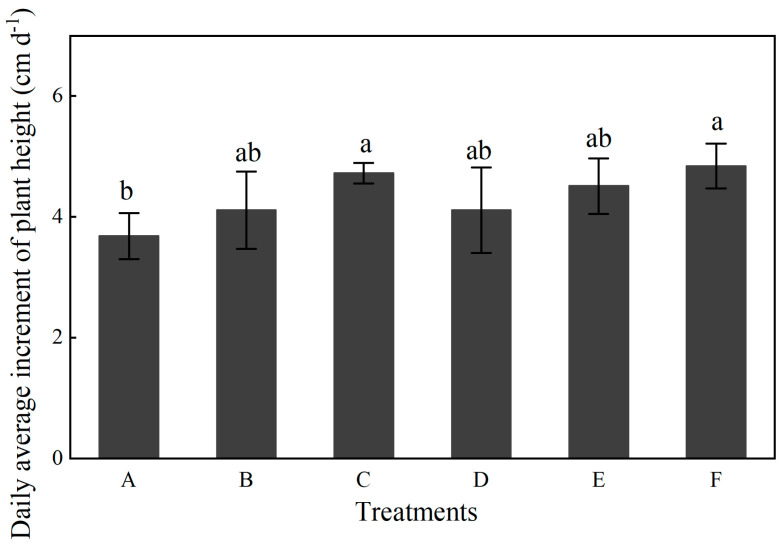
Effects of different treatments on the daily average increase in plant height (2012–2014; n = 10). Lowercase letters above bars indicate significant differences at the 0.05 level. The detailed information regarding treatments A, B, C, D, E, and F can be found in Table A1.

**Figure 2 plants-14-03594-f002:**
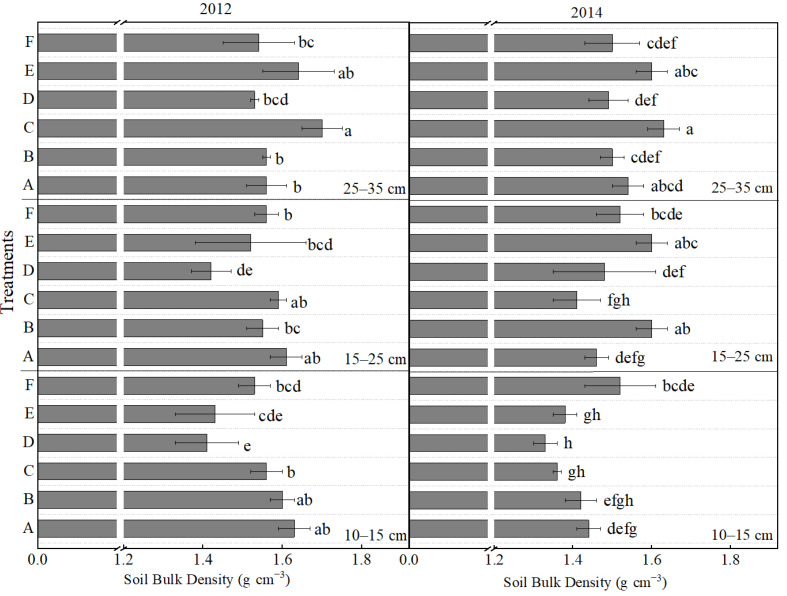
Effects of different treatments on the soil bulk density at different soil layers. Lowercase letters above bars indicate significant differences at the 0.05 level. The detailed information regarding treatments A, B, C, D, E, and F can be found in Table A1.

**Figure 3 plants-14-03594-f003:**
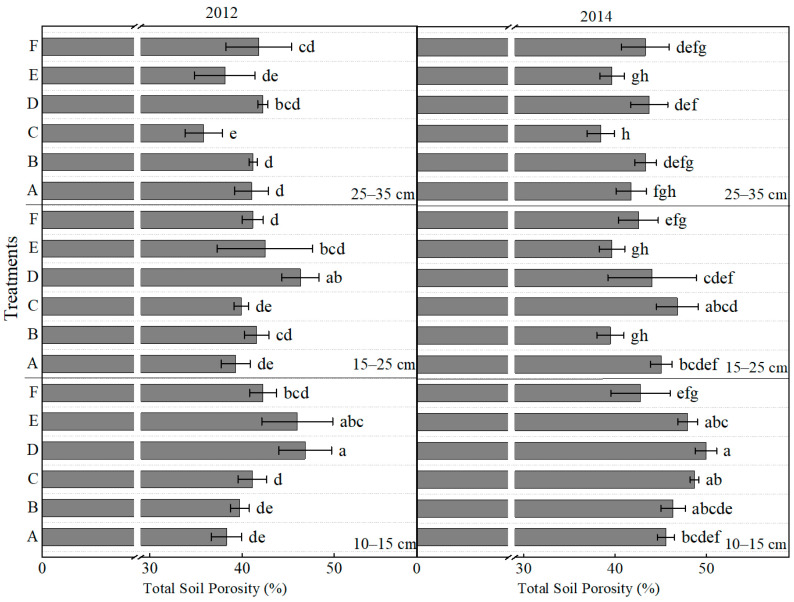
Effects of different treatments on the total soil porosity at different soil layers. Lowercase letters above bars indicate significant differences at the 0.05 level. The detailed information regarding treatments A, B, C, D, E, and F can be found in Table A1.

**Figure 4 plants-14-03594-f004:**
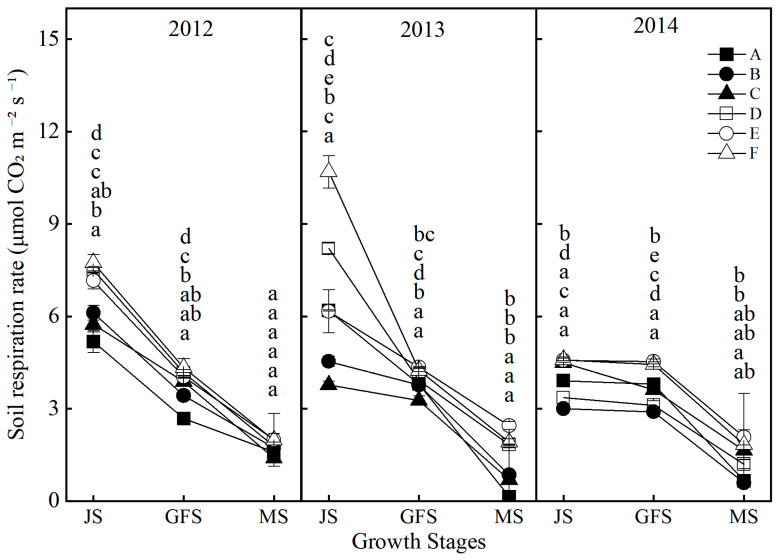
Effects of different treatments on the soil respiration rate at different soil layers. JS, jointing stage. GFS, grain filling stage. MS, maturation stage. Lowercase letters above bars indicate significant differences at the 0.05 level. The detailed information regarding treatments A, B, C, D, E, and F can be found in Table A1.

**Figure 5 plants-14-03594-f005:**
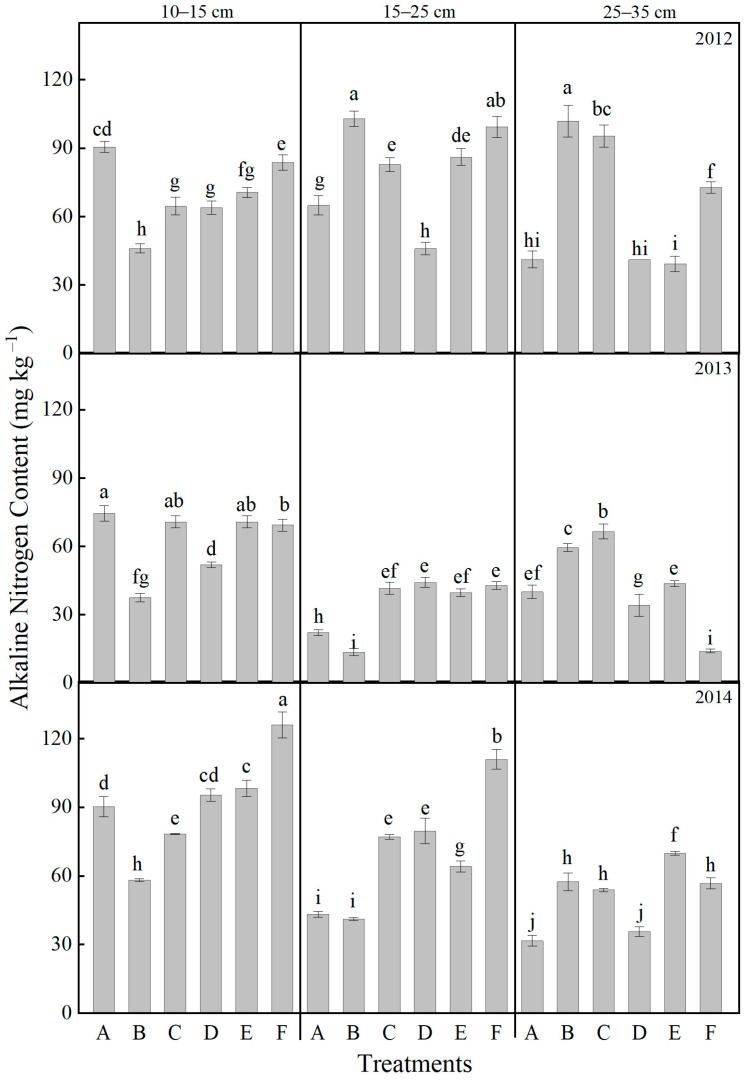
Effects of different treatments on the soil alkaline nitrogen contents at different soil layers. Lowercase letters above bars indicate significant differences at the 0.05 level. The detailed information regarding treatments A, B, C, D, E, and F can be found in Table A1.

**Figure 6 plants-14-03594-f006:**
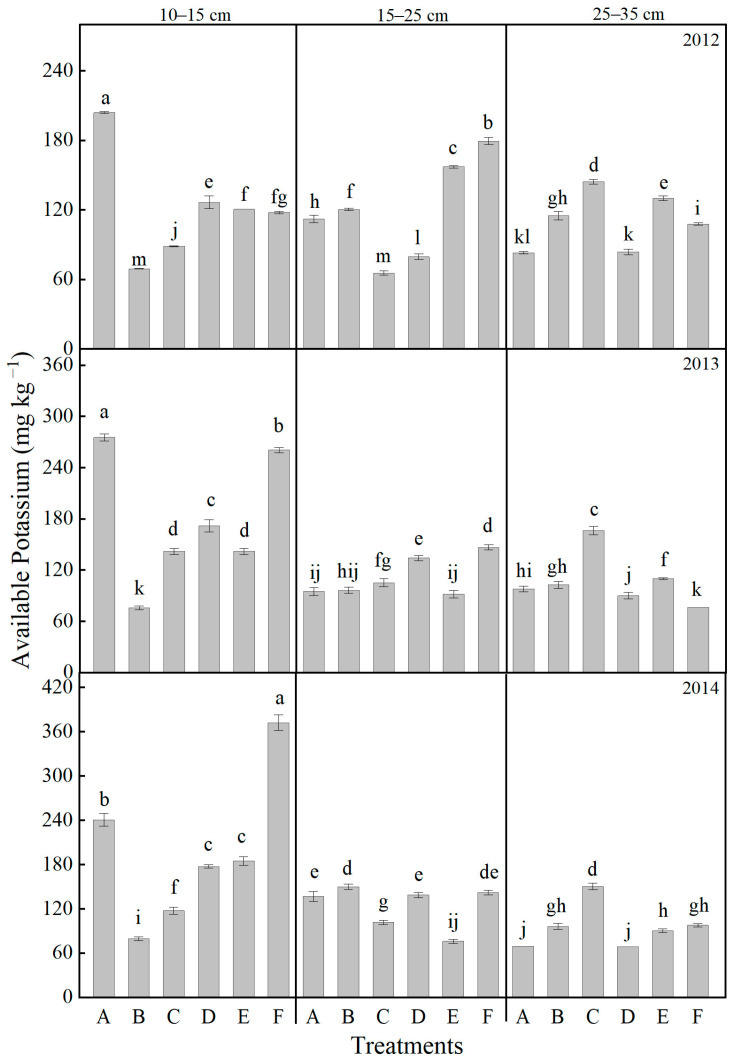
Effects of different treatments on the soil available potassium contents at different soil layers. Lowercase letters above bars indicate significant differences at the 0.05 level. The detailed information regarding treatments A, B, C, D, E, and F can be found in Table A1.

**Figure 7 plants-14-03594-f007:**
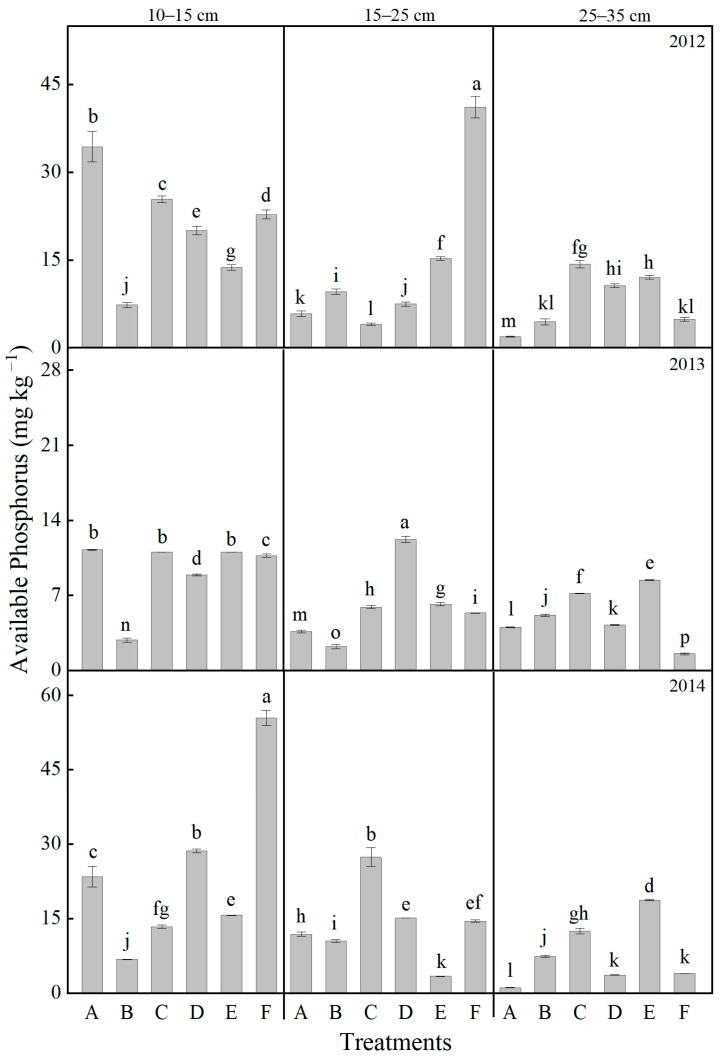
Effects of different treatments on the soil available phosphorus contents at different soil layers. Lowercase letters above bars indicate significant differences at the 0.05 level. The detailed information regarding treatments A, B, C, D, E, and F can be found in Table A1.

**Figure 8 plants-14-03594-f008:**
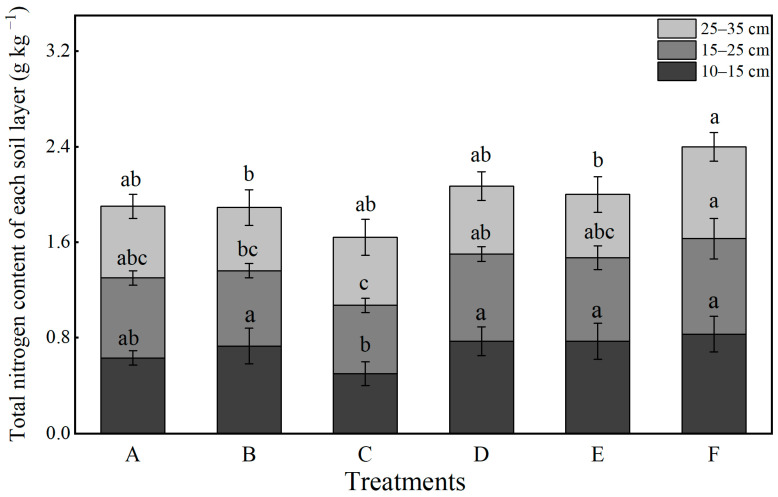
Effects of different treatments on the soil total nitrogen contents at different soil layers (2012–2014). Lowercase letters above bars indicate significant differences at the 0.05 level. The detailed information regarding treatments A, B, C, D, E, and F can be found in Table A1.

**Figure 9 plants-14-03594-f009:**
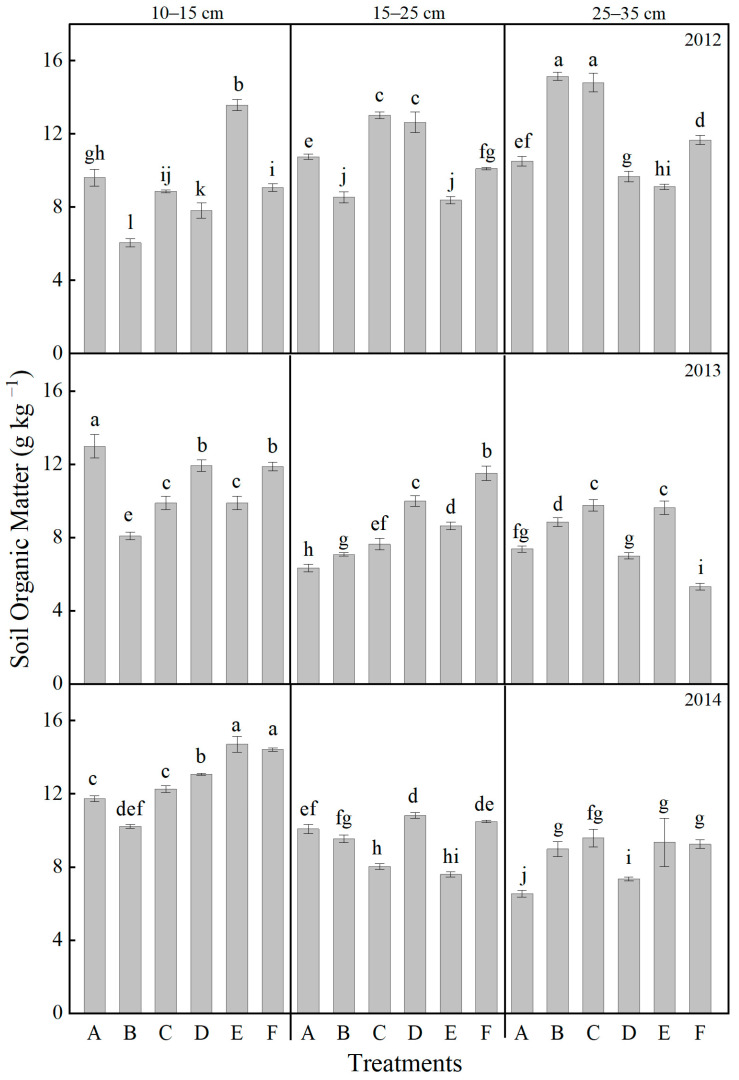
Effects of different treatments on the soil organic matter at different soil layers. Lowercase letters above bars indicate significant differences at the 0.05 level. The detailed information regarding treatments A, B, C, D, E, and F can be found in Table A1.

**Figure 10 plants-14-03594-f010:**
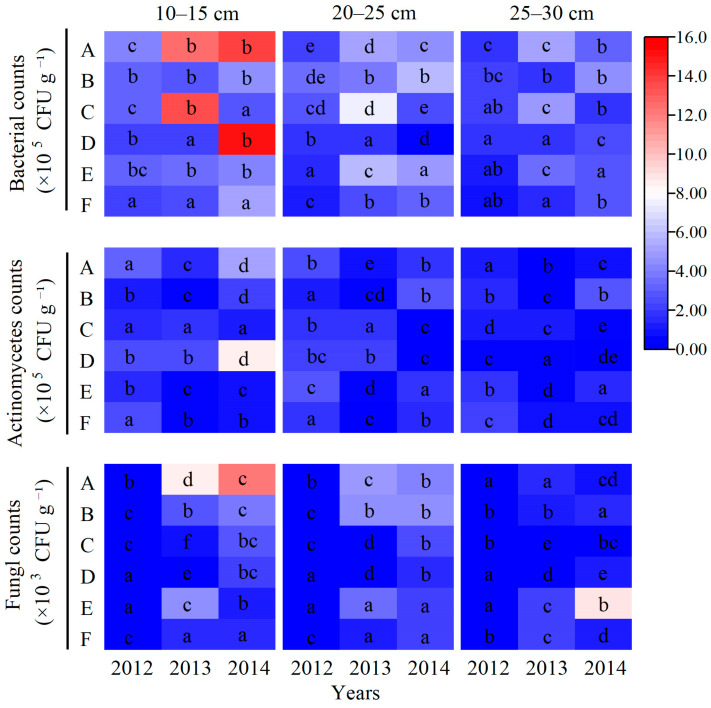
Effects of different treatments on the soil microflora at different soil layers. Different lowercase letters within a single heatmap indicate significant differences at the 0.05 probability level for a given year. The detailed information regarding treatments A, B, C, D, E, and F can be found in Table A1.

**Figure 11 plants-14-03594-f011:**
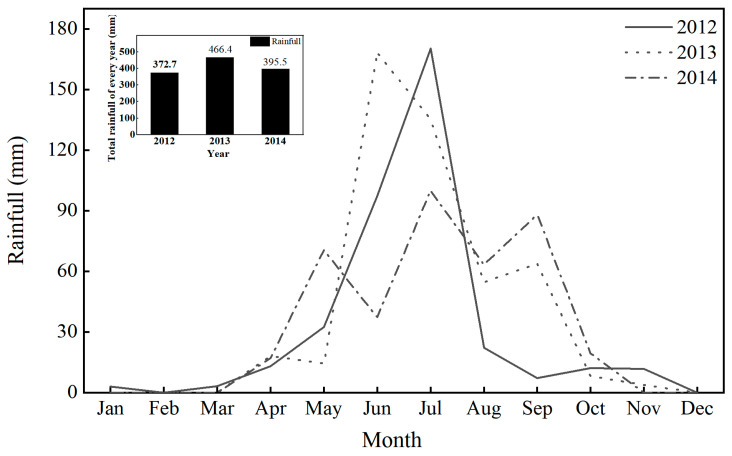
Rainfall distribution at the experimental site from 2012 to 2014.

**Table 1 plants-14-03594-t001:** Effects of different treatments on the plant height, ear height, and stem diameter.

Years	Treatments	Plant Height (cm)		Ear Height (cm)	Stem Diameter (mm)
		Jointing Stage	Tasseling Stage		
2012	A	85.5 a	138.2 d	98.4 c	22.3 ab
	B	76.4 b	149.1 bc	110.6 ab	22.7 ab
	C	78.3 b	156.4 ab	112.7 ab	23.6 ab
	D	78.0 b	154.5 ab	104.1 bc	24.3 a
	E	78.7 b	158.4 a	106.1 abc	23.2 ab
	F	69.6 c	148.2 c	114.1 a	21.3 b
2013	A	57.2 ab	166.8 b	111.0 ab	18.0 b
	B	58.2 a	177.3 a	114.6 ab	19.9 ab
	C	54.7 c	182.5 a	116.7 a	20.3 a
	D	51.7 d	164.8 b	115.0 ab	21.3 a
	E	56.9 ab	178.2 a	107.7 b	19.7 ab
	F	55.9 bc	175.7 a	116.5 a	19.6 ab
2014	A	100.7 a	174.5 c	87.7 cd	21.5 ab
	B	100.1 ab	167.5 c	82.5 d	19.3 c
	C	95.1 bc	185.8 b	95.5 b	22.9 ab
	D	102.7 a	169.9 c	90.7 bc	21.2 abc
	E	101.2 a	182.5 b	82.9 d	20.9 bc
	F	92.7 c	195.8 a	101.3 a	23.0 a
Y		1169.421 **	195.430 **	151.250 **	28.333 **
T		10.225 **	17.183 **	11.780 **	3.188 *
Y × T		4.863 **	6.716 **	2.753 **	2.475 *

Different lowercase letters in a single column indicate significant differences at the 0.05 probability level for a given year. * and ** mean significant differences at the 0.05 and 0.01 probability levels, respectively. Abbreviations: Y, year; T, treatment. The plant height was measured with *n* = 10 replicates, while ear height and stem diameter were assessed with *n* = 7 replicates for each. The detailed information regarding treatments A, B, C, D, E, and F can be found in Table A1.

**Table 2 plants-14-03594-t002:** Effects of different treatments on the maize yield and yield compositions.

Years	Treatments	Ear Length (cm)	Ear Diameter (cm)	Row Number Per Ear	Grain Numbers Per Ear	Ear Tip Barrenness (cm)	100-Grain Weight (g)	Grain Yield (kg ha^−1^)
2012	A	19.5 b	5.2 c	15.3 a	566.9 a	0.63 c	41.7 a	5809 d
	B	19.8 ab	5.3 ab	15.3 a	561.0 a	0.90 ab	41.4 a	6283 c
	C	20.2 ab	5.3 abc	16.7 a	622.5 a	0.80 abc	40.0 b	5887 d
	D	20.4 a	5.4 a	16.0 a	621.2 a	0.80 abc	41.0 ab	6338 c
	E	19.8 ab	5.4 a	16.0 a	579.8 a	0.73 bc	40.8 ab	6855 b
	F	19.5 b	5.2 bc	16.2 a	565.6 a	0.93 a	39.9 b	7479 a
2013	A	18.8 a	5.1 a	17.6 a	565.0 a	0.86 ab	33.0 d	5592 f
	B	19.9 a	5.3 a	16.8 a	621.0 a	0.68 b	36.9 c	6313 e
	C	18.9 a	5.0 a	17.2 a	636.0 a	0.87 ab	35.9 c	7270 d
	D	19.1 a	5.2 a	17.2 a	608.2 a	0.87 ab	41.4 ab	8240 c
	E	17.9 a	5.0 a	17.2 a	636.2 a	0.98 a	40.2 b	8690 b
	F	18.2 a	5.0 a	17.6 a	643.4 a	0.74 ab	42.5 a	8953 a
2014	A	16.6 a	4.6 a	17.2 a	526.8 a	0.88 bc	28.1 d	5994 c
	B	16.1 a	4.6 a	16.8 a	522.0 a	1.12 a	27.0 c	5970 c
	C	17.8 a	4.8 a	18.0 a	570.5 a	0.90 bc	31.7 c	6306 bc
	D	16.6 a	4.8 a	17.0 a	507.0 a	0.82 c	34.3 ab	6639 ab
	E	17.6 a	4.6 a	16.0 a	552.3 a	0.99 b	29.5 b	6792 a
	F	17.4 a	4.7 a	17.0 a	552.2 a	0.97 b	31.6 a	6999 a
Y		13.917 **	62.117 **	6.939 **	9.276	10.268 **	45.242 **	304.106 **
T		0.198	1.530	0.882	1.089	1.619	58.006 **	240.839 **
Y × T		0.497	1.058	0.450	0.513	4.141 **	19.914 **	41.941 **

Different lowercase letters in a single column indicate significant differences at the 0.05 probability level for a given year. ** mean the significant differences at 0.01 probability levels, respectively. Abbreviations: Y, year; T, treatment. The detailed information regarding treatments A, B, C, D, E, and F can be found in Table A1.

**Table 3 plants-14-03594-t003:** Analysis of maize economic evaluation under different treatments.

Treatments	Years	Inputs								Total Inputs	Yield	Unit Price	Yield Income	Net Income	Annual Average Income	Cost–Benefit Ratio
		Ploughing Cost	Rotary Tillage Cost	Sowing Cost	Seed Cost	Mature Compost Cost	Compound Fertilizer Cost	Pesticide Cost	Harvesting Cost							
		CNY ha^−1^	CNY ha^−1^	CNY ha^−1^	CNY ha^−1^	CNY ha^−1^	CNY ha^−1^	CNY ha^−1^	CNY ha^−1^	CNY ha^−1^	Kg ha^−1^	CNY kg^−1^	CNY ha^−1^	CNY ha^−1^	CNY ha^−1^	
A	2012	0	225	150	870	0	3975.0	225	1500	6945.0	5809.1	2.2	12,779.9	5834.9	5810.9	1.8
	2013	0	225	150	870	0	3975.0	225	1500	6945.0	5591.9	2.2	12,302.1	5357.1		
	2014	0	225	150	870	0	3975.0	225	1500	6945.0	5993.6	2.2	13,185.8	6240.8		
B	2012	300	225	150	870	0	3975.0	225	1500	7245.0	6283.1	2.2	13,822.7	6577.7	6369.3	1.9
	2013	300	225	150	870	0	3975.0	225	1500	7245.0	6312.6	2.2	13,887.7	6642.7		
	2014	300	225	150	870	0	3975.0	225	1500	7245.0	5969.3	2.2	13,132.4	5887.4		
C	2012	330	225	150	870	0	3975.0	225	1500	7275.0	5886.9	2.2	12,951.2	5676.2	6997.4	2.0
	2013	330	225	150	870	0	3975.0	225	1500	7275.0	7269.5	2.2	15,992.8	8717.8		
	2014	330	225	150	870	0	3975.0	225	1500	7275.0	6306.0	2.2	13,873.2	6598.2		
D	2012	0	225	150	870	1500	1987.5	225	1500	6457.5	6338.3	2.2	13,944.2	7486.7	9101.6	2.4
	2013	0	225	150	870	1500	1987.5	225	1500	6457.5	8240.0	2.2	18,127.9	11,670.4		
	2014	0	225	150	870	1500	1987.5	225	1500	6457.5	6638.7	2.2	14,605.1	8147.6		
E	2012	0	225	150	870	1500	1987.5	225	1500	6457.5	6854.6	2.2	15,080.0	8622.5	9712.7	2.5
	2013	300	225	150	870	1500	1987.5	225	1500	6757.5	8690.0	2.2	19,117.9	12,360.4		
	2014	330	225	150	870	1500	1987.5	225	1500	6787.5	6792.2	2.2	14,942.7	8155.2		
F	2012	330	225	150	870	1500	1987.5	225	1500	6787.5	7478.7	2.2	16,453.1	9665.6	10,515.2	2.6
	2013	300	225	150	870	1500	1987.5	225	1500	6757.5	8953.1	2.2	19,696.7	12,939.2		
	2014	0	225	150	870	1500	1987.5	225	1500	6457.5	6999.2	2.2	15,398.1	8940.6		

CNY denotes Chinese Yuan per kilogram; all prices presented in the table correspond to the average local prices for the respective year. The detailed information regarding treatments A, B, C, D, E, and F can be found in Table A1.

**Table 4 plants-14-03594-t004:** The base data of the top 35 cm of soil layers before treatments in the experimental site.

Soil Depths (cm)	Total Nitrogen (g kg^−1^)	Available Nitrogen (mg kg^−1^)	Available Phosphorus (mg kg^−1^)	Available Potassium (mg kg^−1^)	Organic Matter (g kg^−1^)	Bulk Density (g cm^−3^)	C/N Ratio
10–15	0.49	61.5	1.73	79.0	6.56	1.54	7.76
15–25	0.61	24.2	3.71	102.0	7.98	1.58	7.59
25–35	0.41	35.7	1.76	82.7	5.24	1.62	7.41

**Table 5 plants-14-03594-t005:** Average nutrient contents of sheep manure compost.

Total Nutrient Content (N + P_2_O_5_ + K_2_O, on Dry Basis) (%)	N (%)	P_2_O_5_ (%)	K_2_O (%)	Organic Matter Content (on Dry Basis) (%)	Moisture Content (Fresh Sample) (%)	Alkali-Hydrolyzable Nitrogen (mg kg^−1^)	Available Potassium (mg kg^−1^)	Available Potassium (mg kg^−1^)
3.2	1.29	0.53	1.34	55.04	3.6	2485	689.6	5068

## Data Availability

The original contributions presented in the study are included in the article; further inquiries can be directed to the corresponding authors.

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
