# Peer review of "Integrated Moldboard Ploughing and Organic–Inorganic Fertilization Enhances Maize Yield and Soil Fertility in a Semi-Arid Region of North China"

_plants, 2025, doi:10.3390/plants14233594_

Round 1

Reviewer 1 Report

Comments and Suggestions for Authors

Introduction

The introduction is informative, but it could be made more concise by focusing on the study's aim in the final paragraph.

Material and methods

Table 4. "30" should be replaced with "35"

line 579 "experimental" should be replaced with design

Results and Discussion

Fig. replace with Figure

line 155 change "were" to "was"

The main theme of your paper is soil–crop interactions under different tillage and fertilisation regimes, not an economic assessment. The “Maize Economic Evaluation” subtitle can be removed as a standalone subtitle; instead, integrate the economic outcome briefly into your Results or Conclusions. Also, subtitle “Economic benefits and regional adaptability of tillage and nutrient management practices”, you can merge it into the Discussion or Conclusion section.

Reviewer 2 Report

Comments and Suggestions for Authors

The research concept is interesting, combining soil tillage at varying depths with organic fertilization. The three-year experiment included a series of extensive studies on maize plants and the soil's physicochemical and biological properties. However, the timing of the study is a drawback – the results cover the years 2012-2014. It is difficult to justify publishing the results more than 10 years after the experiment's completion.

A drawback of the study is the insufficiently detailed description of the experimental sites in terms of the tillage methods used.

Detailed notes:

  • In the abstract, the authors describe three soil tillage methods: ploughing (P), rotary tillage (R), and moldboard ploughing. The methodology lacks information on the precise meaning of these terms and the cultivation tools used. However, the study includes six tillage methods; two each of ploughing and rotary tillage, while objects E and F are actually three treatments each applied in consecutive years, each involving not only moldboard ploughing but also rotary tillage in the first or last year. In summary, the information contained in the abstract does not correspond to the actual scheme, which is more complex.
  • As mentioned above, it is necessary to clarify the difference between the "ploughing" and " moldboard plowing" methods in the experiment. In the literature, the term "ploughing" is also commonly understood as tillage with a moldboard ploughing.
  • Was the experiment conducted in the same location in the following three years (maize monoculture) or was the experiment conducted after other crops (pre-crops)?
  • The results in Table 1 are presented with too much precision. For many years, it has been common practice in the scientific literature to present results with sufficient precision to ensure they are legible to the recipients, i.e., three values, e.g., 100; 10.0; 1.00.
  • A similar comment applies to Table 2, where the yield in kg is given with excessive precision – unnecessary decimal places make it difficult to compare the study results.
  • Figure 1 lacks information about the study years, and Figure 8 is similarly lacking – if this information is included in the manuscript text, it should also be included in the figure caption.
  • The Authors evaluated soil levels 10-15, 15-25, and 25-35 cm. The lack of an assessment of soil properties in the top 0-10 cm layer is surprising. This level is very important in studies on soil tillage systems, especially since the tillage depth used in sites A and D was only 10-15 cm. Therefore, in sites A and D, tillage was limited to only a small extent to the deeper soil layers. This means that the Authors did not evaluated the 0-10 cm layer, which was subjected to tillage in all sites, and often evaluated deeper layers that were not. This is a serious methodological error. It is difficult to agree with the interpretation that research levels 10-15 are considered topsoil, as defined by the authors.
  • In their conclusions, the authors emphasize the beneficial effects on soil properties and maize yield after using moldboard ploughing, while rotary tillage was also used in plots E and F – there was no plot exclusively using moldboard tillage.
  • Of the 83 references, the vast majority have correct editorial notation. Items 28, 32, 35, 44, 49, 53, 64, 68, 71, 72, 76, and 78 require corrections – the article titles are capitalized instead of lowercase.

Reviewer 3 Report

Comments and Suggestions for Authors

Dear All,

This manuscript presents the results of a well-structured three-year field trial evaluating the effects of integrated tillage methods and organic-inorganic fertilization regimes on maize yield, soil fertility, and microbial dynamics in the semi-arid region of northern China. Specifically, the study investigates how different tillage depths (10–15 cm, 15–25 cm, 25–35 cm) and fertilization treatments (inorganic vs. organic-inorganic combinations) affect soil physicochemical properties, nutrient distribution, microbial abundance, and economic returns.

In my view, the main strengths of the study lie in its experimental duration, factorial design, and the multi-layer soil sampling approach that captures vertical distribution of soil parameters. The integration of agronomic, biological, and economic indicators offers a comprehensive evaluation of sustainable farming strategies in dryland agriculture. Treatment F, combining deep moldboard ploughing and compost-compound fertilization, consistently showed the highest yield, microbial activity, and economic benefit - demonstrating clear applicability to sustainable intensification in low-input regions.

Strong Points

  • Multi-year field study capturing interannual variability and environmental response.

  • Systematic evaluation of soil properties at three depths (10–15, 15–25, 25–35 cm).

  • Soil microbial abundance and respiration integrated with agronomic parameters.

  • Practical relevance: findings directly support integrated management in semi-arid systems.

  • Clear link between soil improvement, nutrient stratification, and yield stabilization.

  • Detailed economic assessment strengthens the application potential of the results.

Weaker Aspects – Suggestions and Clarifications

So, I have annotated below an attempt to clarify certain ideas. The authors should examine my suggested changes carefully to ensure that I have not misinterpreted what they wanted to say.

Title and Abstract

  1. Title (Page 1, Line 1)
    “could increase maize yield…” → Consider changing to:
    “enhances maize yield and soil fertility…”
    to reflect certainty based on 3 years of data.

  2. Abstract (Lines 15–31)
    Please avoid passive language and vague phrasing like “could increase.” State clearly what was observed (e.g., “enhanced,” “improved”).

  3. Abstract (Line 21)
    Specify the type of “compound fertilizer” used, e.g., N-P-K ratio 15:15:15 as mentioned later.

  4. Abstract (Lines 25–26)
    Please report microbial increases with units (e.g., CFU g⁻¹) if available or clearly indicate method used.

  5. Abstract (Line 31)
    Consider ending with a stronger conclusion:
    “These findings support the promotion of integrated tillage-fertilization strategies to enhance maize productivity and soil ecological function in semi-arid regions.”

Introduction

  1. Page 2, Line 41
    Reword “suboptimal soil management” → consider:
    “persistent soil degradation due to shallow tillage and excessive chemical inputs” for precision.

  2. Page 2, Line 66
    Add a brief soil classification or reference to support the claim of “complex agro-ecological environments.”

  3. Page 2, Line 70–75
    Cite additional references showing negative effects of overfertilization on water quality or greenhouse gas emissions.

  4. Page 3, Line 97
    Please clarify what is meant by “a novel comprehensive field management strategy” — is this novel within the region or in literature?

Materials and Methods

  1. Page 20, Line 572
    The soil is classified as “brown soil.” Please include WRB classification or USDA equivalent, e.g., Cambisol, Luvisol, etc.

  2. Page 20, Table 4
    Consider reporting C/N ratio of soils pre-treatment.

  3. Page 21, Table 5
    Indicate which treatments were replicated over three years. Treatment E and F appear staggered — clarify rationale for this design.

  4. Page 21, Line 589
    Please define “compound fertilizer” more clearly in the first mention. Include source/manufacturer and NPK details.

  5. Page 22, Line 617
    Please specify soil temperature or range during respiration measurements (9:00–11:00), as it influences microbial activity.

  6. Page 22, Line 626
    Add proper reference for the alkaline hydrolysis method used for soil N determination.

  7. Page 22, Line 638
    Clarify units of microbial enumeration (e.g., CFU g⁻¹ dry soil) and provide limits of detection.

Results and Figures

  1. Figure 1 (Page 5)
    Suggest simplifying bar labels and ensure consistent use of units and legends. Add n = 3 or n = 4 where relevant.

  2. Figure 3 (Page 7)
    Consider adding error bars to show variability in porosity across replicates.

  3. Page 13, Lines 311–319
    Fungal populations are described but not quantified as clearly as bacteria/actinomycetes. Add numerical support if available.

  4. Page 14, Line 339
    Economic assessment is comprehensive but consider adding a brief cost-benefit ratio or ROI metric.

Discussion

  1. Page 16, Line 386–390
    Please clarify if reduced grain filling under chemical-only treatments was measured physiologically or inferred from yield data.

  2. Page 17, Line 455–460
    Consider adding a short paragraph discussing nutrient leaching risk under different stratification profiles.

  3. Page 18, Line 499–503
    The claim of “microbial mediated nutrient mineralization” should be supported by enzyme activity data or reference.

  4. Page 19, Line 544–546
    Please add justification for choosing sheep manure over other organic sources.

  5. Page 19, Line 550–555
    Acknowledge that the single-site design limits generalizability — authors already state this, but it deserves slight expansion.

Language and Style

  1. Several sections (e.g., Pages 4–5) contain redundant phrasing like “significantly increased by 2.3 times” — consider replacing with “increased by 130%” for clarity.

  2. Page 1–4: Frequent use of passive voice; suggest shifting to active where appropriate for impact.

  3. Suggest revising long paragraph blocks in Discussion into smaller subsections with headers, e.g., “Soil Structure”, “Microbial Activity”.

Round 2

Reviewer 2 Report

Comments and Suggestions for Authors

The Authors of the manuscript responded fully to my questions and suggestions. Appropriate additions have been made to the manuscript text. In my opinion, the article in its current form can be recommended for publication in the journal Plants – MDPI.

Author Response

We sincerely thank you for your professional feedback, meticulous attention to detail, and valuable time invested in improving our manuscript. Best wishes for your continued success in research!